# A Targeted Metabolomics Approach to Study Secondary Metabolites and Antioxidant Activity in ‘Kinnow Mandarin’ during Advanced Fruit Maturity

**DOI:** 10.3390/foods11101410

**Published:** 2022-05-13

**Authors:** Manpreet Kaur Saini, Neena Capalash, Eldho Varghese, Charanjit Kaur, Sukhvinder Pal Singh

**Affiliations:** 1Division of Food and Nutritional Biotechnology, National Agri-Food Biotechnology Institute, Mohali 160071, India; saini.manpreetkaur87@gmail.com; 2Department of Biotechnology, Panjab University, Chandigarh 160014, India; caplash@pu.ac.in; 3Fishery Resources Assessment Division, ICAR-Central Marine Fisheries Research Institute, Kochi 682018, India; eldhoiasri@gmail.com; 4Division of Food Science and Post–Harvest Technology, ICAR-Indian Agricultural Research Institute, New Delhi 110012, India; charanjit.kaur@icar.gov.in; 5New South Wales Department of Primary Industries, Ourimbah, NSW 2258, Australia

**Keywords:** antioxidant activity, flavonoids, growing climate, harvest maturity, Kinnow, limonoids, phenolics, metabolomics

## Abstract

In this study, we investigated the impact of harvest maturity stages and contrasting growing climates on secondary metabolites in Kinnow mandarin. Fruit samples were harvested at six harvest maturity stages (M1–M6) from two distinct growing locations falling under subtropical–arid (STA) and subtropical–humid (STH) climates. A high-performance liquid chromatography-tandem mass spectrometry (HPLC-MS/MS) technique was employed to identify and quantify secondary metabolites in the fruit juice. A total of 31 polyphenolics and 4 limonoids, with significant differences (*p* < 0.05) in their concentration, were determined. With advancing maturity, phenolic acids and antioxidant activity were found to increase, whereas flavonoids and limonoids decreased in concentration. There was a transient increase in the concentration of some polyphenolics such as hesperidin, naringin, narirutin, naringenin, neoeriocitrin, rutin, nobiletin and tangeretin, and limonoid aglycones such as limonin and nomilin at mid-maturity stage (M3) which coincided with prevailing low temperature and frost events at growing locations. A higher concentration of limonin and polyphenolics was observed for fruit grown under STH climates in comparison to those grown under STA climates. The data indicate that fruit metabolism during advanced stages of maturation under distinct climatic conditions is fundamental to the flavor, nutrition and processing quality of Kinnow mandarin. This information can help in understanding the optimum maturity stage and preferable climate to source fruits with maximum functional compounds, less bitterness and high consumer acceptability.

## 1. Introduction

Citrus fruits are among the most widely consumed natural products in the world. In recent years, there was an increase in worldwide citrus production and consumption due to an increasing understanding of their health benefits. Citrus fruit draws the attention of researchers due to the presence of metabolites with plentiful health benefits such as antioxidant [1,2], anti-proliferative [3,4,5,6,7], hypocholesterolaemic and anti-diabetic [8,9,10,11], anti-inflammatory [12,13,14,15,16,17] and antiretroviral activities [18,19,20].

The accumulation of these metabolites in citrus is affected by various factors such as species, variety, climate and soil conditions of a location and rootstock and development and maturation [21]. Among these, the harvest maturity and climate of a growing location are strong modulators of secondary metabolites, especially flavonoids, phenolics and limonoids, which ultimately influence consumer preference in the market. Harvest maturity plays a significant role in flavor development in citrus during ripening when there is a decrease in active growth and metabolism shifts to assimilate some physiological and biochemical changes [22]. Various fruit quality traits (color, peelability, seed number, shape, size, texture, SSC, acidity and maturity index) of citrus are acquired during the fruit development and maturation stage, while marked changes in metabolites involved in primary and secondary metabolism occur during the physiological maturity stage [22]. Various studies also described changes in secondary metabolites with maturation in Thai tangerine [23], yuzu mandarins [24], Ponkan and Huoyu mandarin [25], Valencia orange [26], chinotto fruits [27] and Guoqing No. 1 [27]. Environmental conditions and growing locations also influence the accumulation of secondary metabolites [28,29,30,31,32]. The phenolic and flavonoid composition of several citrus species such as sweet orange, yuzu and chinotto during maturity are also reported in the literature [24,27,33,34]. Thus, the harvest maturity and climate of a growing location are considered critical for improving the nutritional value and sensorial attributes of the citrus fruit, as palatability and taste are closely associated with the amount and type of chemical constituents at the time of harvest [35]. Therefore, the basic knowledge and understanding of changes in metabolites during fruit maturation is critical to improve and maintain fruit quality.

Metabolomics has provided a rapid and realistic method for monitoring changes in metabolites in different plants and plant products. Analytical strategies for metabolomics research aimed to characterize the whole metabolite complement and then relate their concentrations to features and properties of the sample [36,37]. Liquid chromatography-tandem mass spectrometry (LC-MS/MS), gas chromatography-mass spectrometry (GC-MS) and nuclear magnetic resonance (NMR) are some of the analytical platforms employed. The analytical results obtained when coupled with chemometrics help in the extraction of valuable information. Various researchers have employed LC-MS and GC-MS-based metabolic profiling to measure metabolic changes associated with fruit development in tomato [38], peach [39], grape [40], citrus [41,42,43] and date palm [44].

Although there are numerous metabolomic studies in citrus, to the best of our knowledge, there is no information on the comprehensive profiling of secondary metabolites at different harvest maturity stages and growing climates in Kinnow mandarin, a hybrid of King tangor (*Citrus nobilis*) and Willow leaf mandarin (*Citrus deliciosa*). Kinnow has become a commercial fruit crop in the Punjab state of India and contributed 1.33 million tonnes in the year 2019-20 [45]. Kinnow mandarin has implicit economic importance due to its high juice content, distinctive flavor, delicious taste and nutritional value position. We reported the impact of different agro-climatic conditions [46] and rootstocks [47] on primary and secondary metabolites in Kinnow mandarin. Results indicated that fruit from subtropical–humid (STH) climatic conditions, and sour orange rootstock favored the accumulation of polyphenolics and limonoids; whereas fruit from subtropical–arid (STA) climatic conditions and rough lemon and Cleopatra mandarin rootstocks favored fruit with lower limonoids and better flavor [46,47]. We previously reported the effect of harvest maturity stages and growing climate on primary metabolites in Kinnow mandarin [48]. Results showed that the concentration of sugars and B-complex vitamins increased, whereas the concentration of organic acids and vitamin C decreased during the final stages of maturation, with a higher concentration of vitamin C in fruit under STH and organic acids in fruit under STA climates [48]. In this paper, we report the effect of harvest maturity stages and growing climate on secondary metabolites and antioxidant activity in Kinnow mandarin, which will complement the published work on primary metabolites [47]. Qualitative and quantitative analysis of secondary metabolites during advanced stages of fruit is helpful in developing appropriate decision tools for harvesting Kinnow according to destination markets and end-use.

## 2. Materials and Methods

### 2.1. Chemicals and Reagents

All the reagents, solvents and reference standards were obtained from Sigma-Aldrich Co. (Bangalore, India) unless mentioned otherwise. HPLC grade formic acid (FA) was obtained from Merck Limited (Mumbai, India). Milli-Q synergy system (Millipore, Bangalore, India) was used for obtaining LC-MS grade water (resistivity 18.2 mΩ). Nomilin and obacunone were procured from ChromaDex (LGC Promochem India Pvt. Ltd., Bangalore, India). All the required reagents for the determination of antioxidant activity were obtained from Analytik Jena (Analytik Jena (AJ) Instruments India Pvt. Ltd., Delhi, India).

### 2.2. Fruit Material and Sampling

Two different locations under contrasting growing climates located in the Punjab state of India were identified for collecting fruit samples: STH—Chhauni Kalan and STA—Abohar (Appendix A). The common rootstock for both the orchards was rough lemon (*C. jambhiri*). The fruit was harvested from each location separately at a regular interval of 10 days, starting from mid-December (M1) to early February (M6). Overall, there were six harvest maturity stages from M1 to M6. At each location, fruit was collected from six selected healthy trees for each maturity stage. Uniform cultural practices were given to all the trees. Ten fruit were harvested from different parts of each tree covering the entire canopy and randomized with fruit samples of other trees. Sixty fruit were harvested from six selected trees at each maturity stage and transported in plastic crates to the laboratory at National Agri-Food Biotechnology Institute (NABI), Mohali, within 6–8 h. The fruit was thoroughly washed to remove adhering dirt and allowed to dry before being peeled off carefully for juice extraction by hand-squeezing. The juice sample was passed through a strainer to remove seeds and fibrous material and then stored in 50 mL-polypropylene tubes at −80 °C until further analyses.

### 2.3. Flavonoids and Phenolics

Samples for polyphenolic analysis were extracted as previously reported by Vrhovsek et al. [49] with minor modifications. Chromatographic conditions, identification and the quantification of polyphenolics were the same as described in previously published papers [45,46,47]. Briefly, thawed Kinnow juice was extracted with methanol: water (80:20; *v*/*v*) and the supernatant obtained after centrifugation was filtered through a 0.2 µm polytetrafluoroethylene (PTFE) membrane filter (Pall India Pvt. Ltd., Mumbai, India) and used for polyphenolics analysis. Zorbax Eclipse plus C–18 column (4.6 mm × 100 mm × 3.5 µm, Agilent Technologies India Pvt. Ltd., New Delhi, India) maintained at 40 °C was used to achieve separation of metabolites using 0.1% formic acid in water as mobile phase A and 0.1% formic acid in acetonitrile as mobile phase B. An Agilent 1260 HPLC coupled to an ion trap mass spectrometer QTRAP^®®^ 5500 (SCIEX, DHR Holdings Pvt. Ltd., Gurgaon, India) was used for identification. Quantification was based on calibration curves of external standards and results were expressed as mg/L.

### 2.4. Limonoid Aglycones and Glycosides

Limonoid aglycones were extracted as described by Manners et al. [50]. Chromatographic conditions, identification and the quantification of limonoid aglycones were conducted as described in previously published papers [46,47]. Briefly, the thawed juice sample was extracted with chloroform, dried using a rotavapor^®®^ R-215 (BUCHI Operations India Private Ltd., Mumbai, India) and reconstituted in 2 mL of acetonitrile. The extracts were filtered using a 0.2 µm PTFE membrane filter and used for the analysis of limonoid aglycones. Aglycones were eluted using 0.1% formic acid in water as mobile phase A and 0.1% formic acid in acetonitrile as mobile phase B.

Limonoid glucosides extraction was carried out following the method described by Jayaprakasha et al. [51]. Chromatographic conditions, identification and the quantification of limonoid aglycones was conducted as described previously in published papers [46,47]. In brief, the juice sample was extracted with methanol followed by centrifugation and the filtered extract was used for analysis. The elution system for glycosides used: solvent A, water: acetonitrile (95: 5, *v*/*v*) and solvent B and water: acetonitrile (5: 95, *v*/*v*). Zorbax Eclipse Plus C-18 column maintained at 35 °C was used for the separation of both limonoid aglycones and limonoids glucosides. The identification of limonoids (aglycones and glucosides) was achieved by coupling an Agilent HPLC system with a QTRAP^®®^ 5500. Calibration curves based on corresponding external standards were used for quantification, and results were expressed as mg/L.

### 2.5. MS/MS Conditions

The MS/MS conditions used were as reported in previously published papers [46,47]. The MS instrument was operated in either positive or negative ESI mode with multiple reaction monitoring (MRM) acquisition. The optimized conditions for the ESI source included: ion spray voltage, 5500 V in positive ion mode and −4500 V in negative ion mode; heater gas (GS2) and nebulizer gas (GS1), 50 psi; curtain (CUR) gas, 30 psi; ion source temperature, 550 °C; the interface heater was on, and collision activated dissociation (CAD) gas, medium. The compound dependent MRM parameters (declustering potential (DP), entrance potential (EP), collision energy (CE) and cell exit potential (CXP)) for each metabolite (precursor-to-product ion transition) are given in Appendix A. The dwell time for each MRM transition was 200 ms.

### 2.6. Limit of Detection (LOD), Limit of Quantification (LOQ), Correlation Coefficient (r) and Calibration Range

The limits of detection (LOD) and quantification (LOQ) for each metabolite were determined from the corresponding calibration curves with LOD = 3 √S/a and LOQ = 10 √S/a (where S is the standard deviation and a is the slope of the calibration curve) (IHT guideline). The LOD, LOQ, r and calibration range for the metabolites are shown in Appendix A.

### 2.7. Antioxidant Activity

The antioxidant activity of the juice sample was determined by a photochemiluminescence (PCL) assay using the Photochem^®^ (Analytik Jena AG, Jena, Germany) as described previously [46] and expressed as activity due to water-soluble (ACW) and lipid-soluble compounds (ACL). Juice samples from the deep freeze (−80 °C) were thawed at ∼22 °C, sonicated in an ultrasonic water bath for 15 min and centrifuged at 10,000 rpm for 5 min at 4 °C. Samples were diluted with reagents from ACW and ACL kits. A comparison of the antioxidant ability of the samples to Trolox and ascorbic acid standards was conducted, and results were expressed as µmol/L. Ascorbic acid and Trolox were taken as positive controls for ACW and ACL, respectively.

### 2.8. Statistical and Multivariate Analysis

The statistical and multivariate analysis procedure was the same as described in a previously published paper for primary metabolites [48]. All the measurements were conducted in triplicate. One biological replicate consisted of 20 hand-squeezed fruits. Two-way analysis of variance (ANOVA) using SAS 9.4 (SAS Institute, Cary, NC, USA) was carried out, and significant effects (*p* < 0.05) were noted. Tukey’s Honest Significant Difference (HSD) test was conducted for pair-wise comparison of the maturity stages, different locations and their interaction. Mean values along with Standard Error of Mean (SEM) and significantly different effects/means (represented by different alphabets) are presented in Table 1. The validity of statistical analysis was ensured by checking all the assumptions of ANOVA.

For multivariate analysis, centered data were log_e_ transformed to reduce heteroscedasticity followed by principal component analysis (PCA) using MarkerView^™^ software (1.2.1, SCIEX). Heap map based on quantitative changes of secondary metabolites was made using Multi Experiment Viewer (MeV) software (version 4.8.1; Dana-Farber Cancer Institute, Boston, MA, USA).

## 3. Results and Discussion

### 3.1. Flavonoids and Phenolics

The major flavonoids and phenolics identified and quantified by LC-MS/MS using MRM analysis are summarized in Table 1. The retention times, molecular formula, quantitative transition (*m*/*z*), MS/MS fragments, collision energy and declustering potential used for identification and quantification are given in the Appendix A. The identification of the phenolic compounds was conducted by comparing both retention times and MS spectral data from both samples and standards. The TIC and mass spectra of polyphenolics analyzed in positive and negative ionization modes are given in the Appendix A. A total of 31 polyphenolics were identified in Kinnow at six harvest maturity stages from M1 (mid–December) to M6 (early–February) and across two growing climates. These compounds were grouped according to different chemical families, namely, flavonoids [flavonones (62.7–69.1%), flavonols (28.6–34.9%), flavones (0.5–0.7%), flavan–3–ols (0.1–0.4%) and isoflavones (traces)], benzoic and cinnamic acid derivatives (1.0–2.8%) and dihydrochalcones (traces). Total polyphenolics in Kinnow showed a decrease in concentration with harvest maturity stages (M1 to M6) from 309.1 to 226.0 mg/L under STA and from 310.1 to 267.2 mg/L under the STH growing climate. The flavanones constituted the most abundant flavonoids in Kinnow and decreased with harvest maturity stages under both STA and STH growing climates. Naringin and neoeriocitrin are examples of Hesperidin was identified as the major flavanone contributing to 17.35–26.59% of the total polyphenolics, followed by naringin (11.37–16.01%), naringenin (13.26–19.90%), narirutin (13.26–18.46%) and neoeriocitrin (0.13–0.81%). neohesperidosides, while hesperidin and narirutin are examples of rutinosides [52]. Hesperidin is generally the most abundant non-bitter bioflavonoid found in tangerines, oranges and lemon, whereas naringin is bitter and found only in pummelo and rough lime but not detected in lime and mandarin orange [53]. However, considerable naringin identified in Kinnow could be attributed to the parentage effect emanating from pummelo. Kinnow is a cross between King (*Citrus nobilis*) × Willow Leaf (*Citrus deliciosa*), the former parent being a cross between mandarin and pummelo. The second most abundant flavonoids identified were flavonols which are known to be free radical scavengers and powerful antioxidants [54]. Flavonols in Kinnow also decreased with harvest maturity (M1 to M6) under both STA and STH growing climates. Major flavonols in Kinnow included rutin hydrate and quercetin, whereas nobiletin, quercetin-3-o-galactoside, tangeretin and kaempferol were found in lower concentrations. The concentration reported is in agreement with previous reports in different citrus species and cultivars [55,56,57]. There was a sharp increase in the concentration of some flavonoids such as hesperidin, naringin, narirutin, naringenin, neoeriocitrin, rutin, nobiletin and tangeretin at the mid-maturity stage (M3).

In addition to flavonoids, phenolics in Kinnow constituted benzoic and cinnamic acid derivatives and dihydrochalcones. Sinapic acid was the most abundant benzoic and cinnamic acid derivatives (phenolic acid) in Kinnow, and average values increased with harvest maturity (M1 to M6) from 2.1 to 3.3 mg/L under STA and 1.8 to 2.2 mg/L under STH growing climates. The other compounds in this group included ferulic, ellagic, syringic, p–coumaric acid, t–cinnamic acid, chlorogenic acid, benzoic acid and caffeic acid. Generally, ferulic acid is abundant in mandarins, but sinapic acid is more abundant in Kinnow, which could be attributed to the parentage of one of its parents King mandarin, to Pummelo [58]. The values are in accordance with previous reports on citrus species such as Ponkan, Huoyu and oranges [59,60]. Total polyphenolics were in higher concentration in Kinnow from STH as compared to those from the STA growing climate (Table 1). The magnitude of decrease of flavonoids with harvest maturity was higher in Kinnow under STA as compared to those under the STH growing climate.

The total flavonoids and phenolic concentration appeared to follow a predictable pattern over fruit maturity, occurring at the highest levels at the initial stages and declining with maturity. However, a sharp increase was observed in the concentration of some flavonoids (e.g., hesperidin, naringin, narirutin, naringenin, neoeriocitrin, rutin, nobiletin and tangeretin) at mid-maturity stage (M3) in Kinnow which coincided with prevailing low temperatures and frost events at growing locations. The declining trend in flavonoids is in agreement with the trend reported in red grapefruit [61], sweet oranges [62], *Citrus limon* [63] and Ponkan [59]. Citrus flavonoids are at their peak in young fruits, as maximum synthesis occurs during the initial stages of fruit growth [64]. However, during cell growth and development, the dilution effect caused by increased water content in juice vesicles could be responsible for the decrease in flavonoids [65]. Flavonoids are differentially regulated at developmental stages, and in general, there is a high expression of phenylalanine lyase (PAL), chalcone synthase (CHS) and chalcone isomerase (CHI) in immature fruits than in mature ones [61]. PAL is the first and main flux point for flavonoid synthesis and the phenylpropanoid pathway and also seems to be a rate-limiting flux point for other phenolic secondary metabolites. Previous gene expression studies in *Citrus Unshiu* Marc indicated a higher accumulation of flavonoids and expression of flavonoid biosynthetic genes in young fruits suggesting the synthesis of flavonoids in the initial developmental stage [66]. Citrus is a non-climacteric fruit; its’ ripening to maturity is regulated by abscisic acid. The production of secondary metabolites in plants is slow as the energy and substrate are directed into growth and maturation processes. Flavones are activated with changes in cellular redox homeostasis and enhanced light. This might lead to the inactivation of antioxidant enzymes and upregulation of flavonols biosynthesis [67]. The induction of transcriptome modifications leading to the enhancement of the flavonoid biosynthesis pathway in blood oranges under conditions of cold stress was also reported [68]. Zhang et al. [32] reported high naringin flavanone glycoside accumulation in pummelo grown under subtropical monsoon climates in comparison to arid conditions. Higher polyphenolics observed in Kinnow from STH as compared to STA growing climate could be attributed to the strong influence of climatic conditions such as temperature, rainfall and solar radiation on flavonoid metabolism [67], while lower polyphenolics in arid climates could be attributed to an increase in temperature leading to a reduction in plant secondary metabolites [7].

### 3.2. Limonoids

Limonoids, the predominant triterpenoids responsible for delayed bitterness in citrus, are known to follow a dynamic pattern during fruit development and maturity [69]. Limonoids in Kinnow included limonoid aglycones (limonin, nomilin, obacunone) and limonoid glucosides (limonin glucoside). The retention times, molecular formula, quantitative transition (*m*/*z*), MS/MS fragments, collision energy and declustering potential used for the identification and quantification of limonoids are given in the Appendix A. The TIC and mass spectra of limonoids analyzed in positive and negative ionization modes are also given in Appendix A. With progressive maturity, there was a decline in limonoid aglycones, with significant variation across both the growing climates (Figure 1a), while the reverse trend was observed in the case of limonin glucoside (Figure 1b). Limonin concentration in Kinnow under the STA growing climate decreased from 4.13–3.55 mg/L at M1–M2, showing an abrupt increase to 10.54 mg/L at M3, followed by a decrease in concentration to 4.41 mg/L at M6 (Figure 1a). Similarly, limonin concentration in Kinnow under the STH growing climate decreased from 6.91–4.70 mg/L at M1–M2, followed by an abrupt increase to 6.78 mg/L at M3 and finally decreasing to 4.97 mg/L at M6 (Figure 1a). A similar decrease in limonin levels with maturity was reported in Valencia oranges and Thai tangerines [23,26,70]. Changes in limonin with maturation can be related to the dilution that occurs due to an increase in fruit size and accumulation of water [28]. A sharp increase in limonoids was observed during the M3 stage at both STA and STH growing climates coinciding with a low temperature and frost event during the winter months. Low temperature can significantly induce limonin accumulation due to cold stress. Unusual weather, temperature fluctuations and harvesting conditions causing the disruption of fruit tissues can promote acidic pH and enzyme activity, promoting the conversion of limonoate A-ring lactone to limonin, thus increasing the bitterness of juice [71]. Other limonoid aglycones (nomilin and obacunone) were detected in traces (<0.01 mg/L). Higher limonin concentration was observed at initial harvest maturity stages M1–M2 in Kinnow under STH as compared to those under the STA growing climate. High limonin content at STH climatic conditions could be attributed to the combined effect of temperature fluctuations and suppressing the action of sugars, citric acid and naringin [71]. Limonin glucoside concentration increased with harvest maturity across both STA (75.56–88.21 mg/L) and STH (47.08–66.72 mg/L) growing climates (Figure 1b). A sharp decrease in limonin glucoside concentration was observed during the M3 stage in Kinnow under both STA (68.13 mg/L) and STH growing climates (44.53 mg/L), coinciding with a sharp increase in limonin concentration at the M3 stage (Figure 1b).

Studies in Thai tangerines [23] and Valencia oranges [70] also reported a decline in limonin levels with maturity. It was reported that an accumulation of limonoid occurs at the initial stages of fruit maturation and then decreases from the middle to late stage of fruit development as the degradation and dilution takes place due to an increase in fruit size and glycosylation of limonoid aglycones. Higher limonoid glucosyltransferase activity encourages the conversion of limonoid aglycones into glucosides leading to a reduction in bitter limonoid aglycones [71]. An abrupt increase in limonin concentration observed at the M3 stage could be attributed to low temperature, which significantly induces limonin accumulation due to cold stress [71]. Higher limonin levels were observed in Kinnow from STH as compared to STA growing climate. Lower limonoids in arid climates could be attributed to the elevation of temperature, causing a reduction in the plants’ secondary metabolites [72].

### 3.3. Antioxidant Activity

The total antioxidant activity (AOX) is expressed as the sum of antioxidant activity due to lipid-soluble (ACL) and water-soluble compounds (ACW) (Figure 2). Antioxidants such are flavonoids, ascorbic acid, amino acids, etc., are detected in the water-soluble fraction, while tocopherols, tocotrienols, carotenoids, etc., are detected in the lipid-soluble fraction [73]. There was an increase in both ACW and ACL with progressive maturity, which could be attributed to the synergistic effects between different phytochemicals, including naringenin, hesperidin, limonoids, quercetin, vitamin C and carotenoids [74]. However, fruits from STH climatic conditions consistently showed higher AOX than STA climatic conditions at all maturity stages; higher limonoids and phenolics in fruit from STH climatic conditions may account for such difference. The increasing trend in AOX is in line with previous studies in sweet orange [75] and other mandarins [74]. Studies reported possible synergistic effects between different antioxidant components such as vitamin C, flavonoids and limonoids which may cause an increase or decrease in antioxidant activity [76]. Synergistic effects between naringenin, hesperidin, quercetin, vitamin C and carotenoids were also reported in citrus juice [74].

### 3.4. Multivariate Analysis

PCA score plots revealed the overall metabolome pattern of the combined effect of growing climates and six different maturity stages (Figure 3), and a clear-cut distinction could be seen in maturity stages at different growing climates. The principal components 1 (PC1) and 2 (PC2) accounted for 58.9% and 17.0%, respectively, of the total data variance in metabolites in fruit from six maturity stages and two growing climates (Figure 4). The first two PCs collectively could explain 75.9% of the total variance. In general, PCA on the basis of metabolites could discriminate the impact of two growing climates by clustering fruit samples together from the STH location and those from STA.

In addition, individual PCA score plots based on harvest maturity stages under each climatic condition were plotted. Principal components 1 (PC1) and 2 (PC2) accounted for 72.9% and 15.1% of the data variance in Kinnow with harvest maturity under the STA growing climate (Figure 5) and 59.1% and 14.8% of the data variance in Kinnow with harvest maturity under the STH growing climate (Figure 6). The six different maturity stages under each growing climate were segregated based on the metabolic pattern. The first two PCs collectively could explain 88.0% of the total variance under STA and 73.9% of the total variance under STH growing climates. The comparative quantitative changes in the flavonoids, phenolics and limonoids of Kinnow from two growing climates and six maturity stages are shown in the heat maps (Figure 6).

## 4. Conclusions

To the best of our knowledge, this is the first report on the characterization of flavonoids, phenolics and limonoids in Kinnow as affected by harvest maturity and growing climates. With advancing maturity, flavonoids and limonoids were found to decrease. However, there was an increase in phenolics and antioxidant activity as harvest maturity progressed. There was a transient increase in the content of some polyphenolics (hesperidin, naringin, narirutin, naringenin, neoeriocitrin, rutin, nobiletin and tangeretin), ascorbic acid, limonin and nomilin at the mid–maturity stage which coincided with prevailing low temperature and frost events. Kinnow from the STA growing climate with lower concentrations of limonin and optimum polyphenolics, and maturity stage M2 with low limonin and optimum polyphenolics seemed to be better suited for the fresh and processing industry. The information generated might help to identify maturity markers to determine the optimum maturity stage in Kinnow. Furthermore, the knowledge gained from this study helps to understand the optimum harvest maturity stage and preferable climatic conditions to source fruits with maximum functional compounds, less bitterness and high consumer acceptability.

## Figures and Tables

**Figure 1 foods-11-01410-f001:**
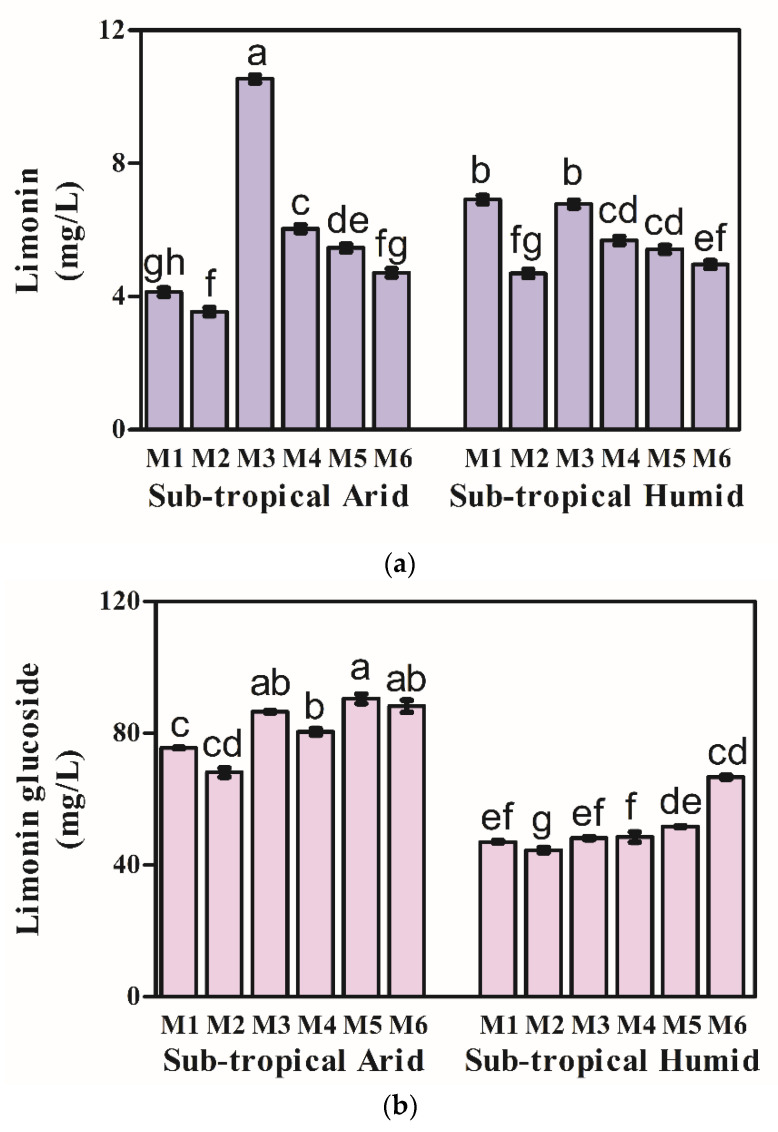
Metabolic profile of (**a**) limonin and (**b**) limonin glucoside in Kinnow as the function of maturity stages and growing climate. Measurements were made in triplicates (*n* = 3; 20 Kinnows/replicate); Vertical bars represent mean values along with the standard error of the mean; Vertical bars with different superscripts are significantly different (*p* < 0.05); M1 to M6 represent six maturity stages (M1–M2: mid and late December, M3–M5: early, mid and late January and M6: early February).

**Figure 2 foods-11-01410-f002:**
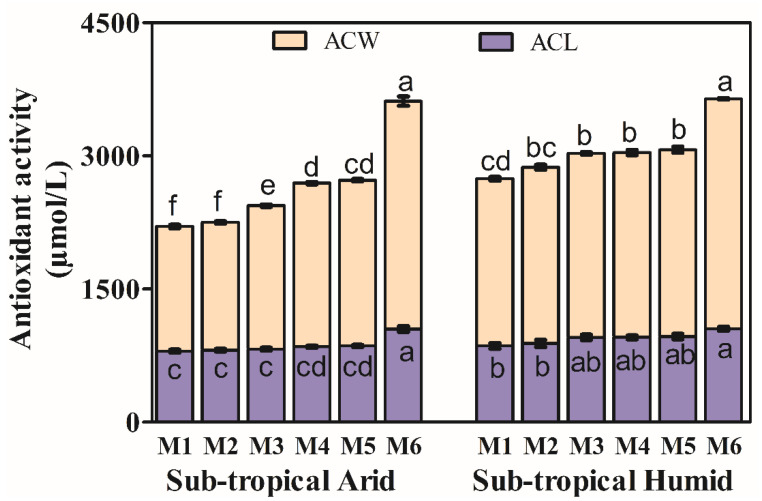
Antioxidant activity in Kinnow as a function of maturity stages and growing climates. Measurements were made in triplicates (*n* = 3; 20 Kinnows/replicate); Vertical bars represent mean values along with the standard error of the mean; Vertical bars with different superscripts are significantly different (*p* < 0.05); M1 to M6 represent six maturity stages (M1–M2: mid and late December, M3–M5: early, mid and late January and M6: early February).

**Figure 3 foods-11-01410-f003:**
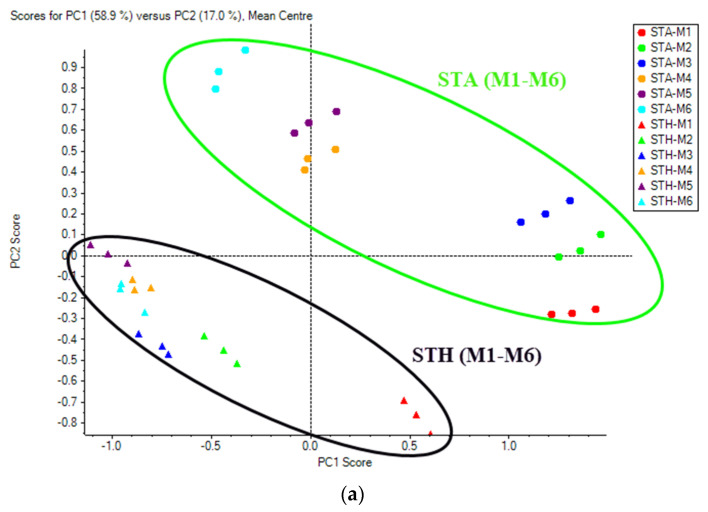
(**a**) PCA score plots and (**b**) loading plots of secondary metabolites in Kinnow as a combined effect of maturity stages and growing climates (STA and STH).

**Figure 4 foods-11-01410-f004:**
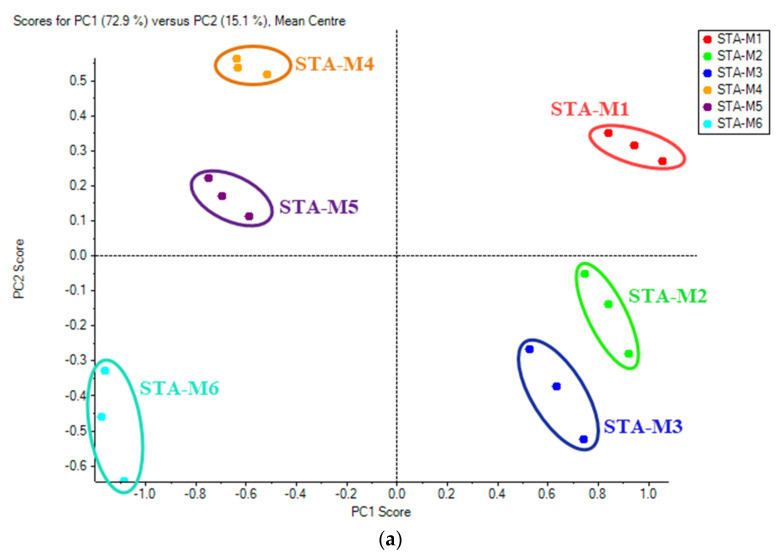
(**a**) PCA score plots and (**b**) loading plots of secondary metabolites at different maturity stages in Kinnow under the STA growing climate.

**Figure 5 foods-11-01410-f005:**
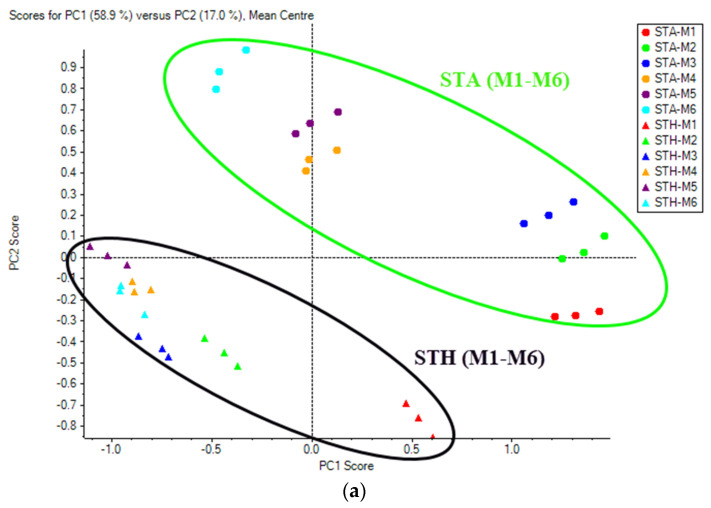
(**a**) PCA score plots and (**b**) loading plots of secondary metabolites at different maturity stages in Kinnow under STH growing climate.

**Figure 6 foods-11-01410-f006:**
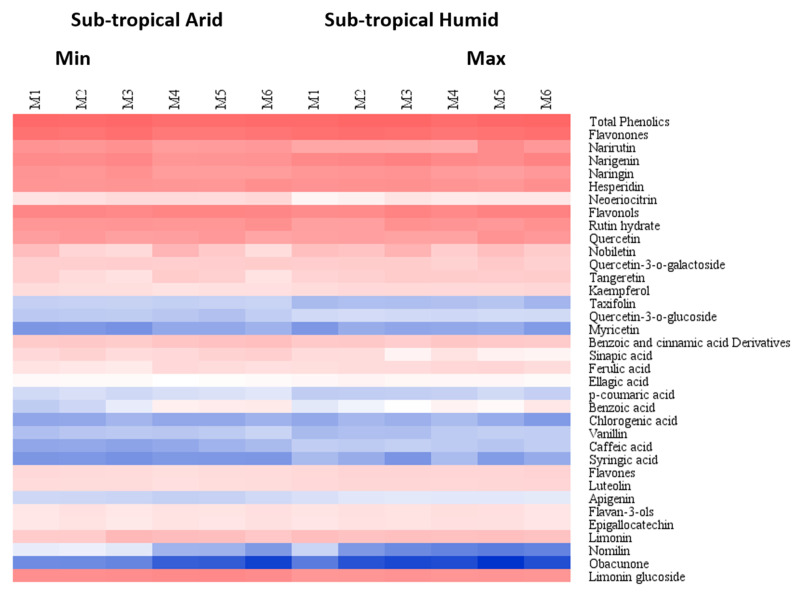
Heat map of secondary metabolites in Kinnow at different maturity stages from STA and STH growing climates.

**Table 1 foods-11-01410-t001:** Metabolic profiles of polyphenolics (mg/L) in Kinnow mandarin as the function of maturity stages and growing climate.

Growing Climate (C)	Subtropical Arid (STA)						Subtropical Humid (STH)						S.E.M
Harvest Maturity (M)	M1	M2	M3	M4	M5	M6	M1	M2	M3	M4	M5	M6	
**Total Phenolics**	309.1 ^d^	274.1 ^fg^	375.5 ^b^	266.9 ^g^	264.7 ^g^	226.0 ^h^	310.1 ^d^	288.8 ^ef^	414.1 ^a^	327.4 ^c^	294.7 ^de^	267.2 ^g^	3.34
**Flavonones**	213.4 ^c^	180.0 ^fg^	259.5 ^b^	170.0 ^g^	177.9 ^fg^	152.3 ^h^	197.6 ^de^	181.1 ^fg^	275.6 ^a^	206.4 ^cd^	190.8 ^ef^	172.1 ^g^	2.56
Hesperidin	70.6 ^c^	62.3 ^d^	94.2 ^b^	48.9 ^fg^	43.9 ^gh^	39.8 ^h^	54.2 ^ef^	50.1 ^efg^	110.1 ^a^	72.5 ^c^	56.7 ^de^	54.3 ^ef^	1.53
Naringin	47.3 ^bc^	40.0 ^d^	60.11 ^a^	32.1 ^e^	30.1 ^e^	29.6 ^e^	52.2 ^b^	45.7 ^c^	50.7 ^bc^	31.5 ^e^	34.9 ^de^	33.9 ^e^	1.07
Narirutin	49.9 ^bc^	41.0 ^de^	62.2 ^a^	49.9 ^bc^	35.1 ^ef^	30.3 ^f^	46.1 ^cd^	46.5 ^cd^	56.1 ^ab^	56.5 ^ab^	54.4 ^b^	42.6 ^cde^	1.51
Naringenin	44.3 ^b^	41.2 ^b^	56.6 ^a^	53.1 ^a^	40.3 ^b^	39.9 ^b^	44.7 ^b^	38.3 ^b^	57.5 ^a^	45.1 ^b^	44.0 ^b^	40.4 ^b^	1.51
Neoeriocitrin	1.5 ^c^	1.33 ^c^	2.3 ^a^	2.16 ^a^	1.80 ^b^	1.8 ^b^	0.4 ^e^	0.5 ^e^	1.2 ^c^	0.8 ^d^	0.8 ^d^	0.8 ^d^	0.06
**Flavonols**	88.3 ^f^	86.7 ^f^	108.7 ^bc^	88.5 ^f^	78.3 ^g^	64.8 ^h^	105.1 ^cd^	100.7 ^de^	131.5 ^a^	113.9 ^b^	96.4 ^e^	87.0 ^f^	1.36
Quercetin	30.9 ^de^	35.6 ^d^	36.0 ^cd^	38.6 ^cd^	30.9 ^de^	30.5 ^de^	45.3 ^b^	43.0 ^b^	55.3 ^a^	53.0 ^a^	42.5 ^b^	43.9 ^b^	1.30
Rutin hydrate	41.9 ^cd^	39.9 ^d^	53.5 ^a^	51.2 ^ab^	39.4 ^d^	37.8 ^de^	46.2 ^bc^	46.4 ^bc^	54.5 ^a^	46.7 ^bc^	43.3 ^cd^	33.1 ^e^	1.20
Nobiletin	7.3 ^c^	3.9 ^g^	9.8 ^b^	2.2 ^i^	2.0 ^i^	1.4 ^j^	6.3 ^d^	4.6 ^f^	11.8 ^a^	5.7 ^e^	3.1 ^h^	3.0 ^h^	0.09
Tangeretin	3.2 ^b^	2.7 ^cd^	3.4 ^b^	1.5 ^e^	1.4 ^e^	1.0 ^f^	3.3 ^b^	2.8 ^c^	3.7 ^a^	3.3 ^b^	2.5 ^d^	2.5 ^d^	0.05
Quercetin-3-O-galactoside	3.3 ^bcd^	3.0 ^cde^	4.0 ^a^	3.5 ^b^	3.1 ^bcde^	2.9 ^de^	2.4 ^fg^	2.3 ^g^	4.0 ^a^	3.3 ^bc^	3.1 ^cde^	2.8 ^ef^	0.08
Kaempferol	1.5 ^c^	1.3 ^de^	1.7 ^b^	1.3 ^e^	1.2 ^f^	1.0 ^g^	1.5 ^c^	1.4 ^cd^	2.1 ^a^	1.7 ^b^	1.7 ^b^	1.5 ^c^	0.02
**Flavan-3-ols**	1.2 ^ab^	1.1 ^ab^	1.4 ^ab^	1.2 ^ab^	1.1 ^ab^	1.0 ^ab^	1.0 ^ab^	1.1 ^ab^	1.4 ^a^	1.3 ^ab^	1.2 ^ab^	0.9 ^b^	0.09
Epigallocatechin	1.2 ^ab^	1.1 ^ab^	1.4 ^ab^	1.2 ^ab^	1.1 ^ab^	1.0 ^ab^	1.0 ^ab^	1.1 ^ab^	1.4 ^a^	1.3 ^ab^	1.2 ^ab^	0.9 ^b^	0.09
**Flavones**	1.8 ^cd^	1.6 ^de^	1.9 ^bc^	1.6 ^de^	1.5 ^ef^	1.3 ^f^	1.9 ^bc^	1.8 ^c^	2.4 ^a^	2.1 ^b^	2.1 ^b^	1.9 ^c^	0.04
Luteolin	1.7 ^c^	1.5 ^d^	1.8 ^bc^	1.5 ^d^	1.4 ^e^	1.2 ^f^	1.8 ^c^	1.7 ^c^	2.2 ^a^	1.9 ^b^	1.9 ^b^	1.7 ^c^	0.02
Genistein	0.1 ^a^	0.1 ^a^	0.1 ^a^	0.1 ^a^	0.1 ^a^	0.1 ^a^	0.1 ^a^	0.1 ^a^	0.2 ^a^	0.2 ^a^	0.2 ^a^	0.2 ^a^	0.02
**Benzoic and Cinnamic Acid Derivatives**	4.0 ^de^	4.3 ^cd^	3.7 ^de^	5.3 ^bc^	5.8 ^ab^	6.4 ^a^	4.1 ^de^	3.6 ^e^	3.6 ^e^	4.3 ^cd^	4.9 ^b^	5.9 ^ab^	0.16
Sinapic acid	2.1 ^d^	2.7 ^b^	1.9 ^d^	2.2 ^cd^	2.7 ^bc^	3.3 ^a^	1.8 ^de^	1.4 ^e^	1.4 ^e^	1.4 ^e^	1.5 ^e^	2.2 ^cd^	0.10
Ferulic acid	1.1 ^def^	0.8 ^f^	0.8 ^ef^	1.3 ^bcdef^	1.5 ^abcd^	1.7 ^abc^	1.4 ^abcde^	1.3 ^bcdef^	1.2 ^cdef^	1.7 ^abc^	1.9 ^a^	1.8 ^ab^	0.11
Ellagic acid	0.5 ^abcd^	0.5 ^cde^	0.6 ^ab^	0.5 ^cde^	0.4 ^ef^	0.4 ^f^	0.5 ^abcd^	0.6 ^abc^	0.4 ^ef^	0.5 ^de^	0.5 ^bcd^	0.6 ^a^	0.02
Benzoic acid	0.1 ^e^	0.1 ^e^	0.3 ^d^	1.1 ^a^	1.0 ^a^	0.8 ^b^	0.2 ^de^	0.3 ^d^	0.5 ^c^	0.5 ^c^	0.7 ^b^	1.1 ^a^	0.02

Data are expressed as mean values (average of three replicates); Mean values with different superscripts within the same row are significantly different (*p* < 0.05); S.E.M: Standard error mean; Measurements were made in triplicates (n = 3; 20 fruit/replicate). M1 to M6 represents six maturity stages (M1–M2: mid and late December, M3–M5: early, mid and late January and M6: early February).

## Data Availability

The data presented in this study are available on request from the corresponding author.

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
