# Peer review of "A Targeted Metabolomics Approach to Study Secondary Metabolites and Antioxidant Activity in ‘Kinnow Mandarin’ during Advanced Fruit Maturity"

_foods, 2022, doi:10.3390/foods11101410_

Round 1
Reviewer 1 Report
Ln 40 - 47. Please cite the references to support the statement
Please the authors must define the acronyms the first time that appear in the text (e.i. STH Ln 81, STA Ln 83, PTFE Ln 118...)
Fruit material and sampling: The data presented in this section is incomplete, the authors must specify the cultivation management and soil conditions. In addition, the authors must specify the number of fruits used, the collection methodology, and the way of transforming the fruits into a juice.
Please specify the number of replications of the analyzes
Change units to SI
Improve the quality of figures 4 and 5
Change the colors in Figure 6 to the lighter standard blue and red for easier viewing
Author Response
Comment 1: Ln 40 - 47. Please cite the references to support the statement
Response: We have added appropriate references (Lado et al., 2018 and Iglesias et al., 2007) to support the statement (line 44. 49)
Comment 2: Please the authors must define the acronyms the first time that appear in the text (e.i. STH Ln 81, STA Ln 83, PTFE Ln 118...)
Response: We have defined the acronyms the first time they appear in the text (line 85, 86 and 134).
Comment 3: Fruit material and sampling: The data presented in this section is incomplete, the authors must specify the cultivation management and soil conditions. In addition, the authors must specify the number of fruits used, the collection methodology, and the way of transforming the fruits into a juice.
Response: We have described in detail the fruit material and sampling in our previous paper on harvest maturity stages and growing climate on primary metabolites in Kinnow mandarin and added reference for the same. Agro-climatic conditions were described in supplementary table S1. However, as per your suggestion we have now added the details for fruit material and sampling in the manuscript itself (line 116).
Comment 4: Please specify the number of replications of the analyses.
Response: We used three biological replicates for the analysis. Twenty fruit constituted one biological replicate for better understanding. This has been mentioned in statistical analysis section (line 192).
Comment 5: Change units to SI
Response: As suggested, we have changed the units (µg/mL, µmol/100mL, ng/mL) to SI (mg/L, µmol/L, µg/L).
Comment 6: Improve the quality of figures 4 and 5
Response: All the figures have been reworked for better quality and understanding.
Comment 7: Change the colors in Figure 6 to the lighter standard blue and red for easier viewing
Response: The figure has been reworked and improved as per your suggestion.
Reviewer 2 Report
Title: Targeted metabolomics approach to study secondary me-tabolites and antioxidant capacity in ‘Kinnow mandarin’ during advanced fruit maturity
Foods
Manuscript Number: foods-1667397
The manuscript is well written and describes interesting study. In my opinion, there are some issues that should be corrected and clarified.
The title should be revised. Please consider the use of antioxidant capacity. You are not analysing that. You just use simple antioxidant assay to evaluate the activity not going deep to mechanism or in vivo antioxidant assay. Replace capacity with activity.
Also change me-tabolites with metabolites
No positive control was presented for antioxidant activity
Please provide the correlation between antioxidant activity and metabolites to show the chemical markers
The authors need to mention why they did targeted metabolomics study
Pg 1, Line 21-22; There was a sharp increase in the concentration of phenolics, limonin and nomilin at mid-maturity stage which coincided with prevailing low temperature and frost events. Please revise the sentence by mentioning the phenolics that were increased. We cannot see all phenolics are significantly higher in mid maturation stage.
Also the same observation is noted in loading plot. Some phenolics are not contributing to the mid stage. Please revise the result and discuss properly as well as justify it.
Pg 1, Line 36-39; Citrus fruit have drawn attention of re-36 searchers due to the presence of metabolites with plentiful health benefits such as antiox-idant activity [1,2], anti-proliferative activity [3–7], hypocholesterolaemic and anti-diabetic activity [8–11], anti-inflammatory activity [12–17], and antiretroviral activity [18–20]. Please revise the sentence and avoid the repetition of activity
Pg 1, line 41; Accumulation of these metabolites in citrus is affected by various factors such as species, variety, climatic and soil conditions of a location, rootstock, and development and maturation. Please revise the sentence (many and) and replace climatic with climate
Pg 2, line 79; please remove position or check the sentence for suitable word choice
Please revise the statistical analysis in Table 1. The alphabets are not correctly represented. You have 2 factors. It is better to represent each one of them. Maybe you can put capital and letters. Measurements were made in triplicates (n = 3; 20 Kinnows/replicate). It is confusing. Please revise it
The Figures are not clear and are too small, especially 3, 4 and 5. Please make them clear and increase their resolutions. Also check the statistics for figures
The titles of figures should be revised. The labelling is presented in the figure. No need to repeat it in legend
The conclusion should be rewritten to show the significant findings and answer the objective of the study. please conclude which stage and climate are better
Author Response
Comment 1: The title should be revised. Please consider the use of antioxidant capacity. You are not analysing that. You just use simple antioxidant assay to evaluate the activity not going deep to mechanism or in vivo antioxidant assay. Replace capacity with activity.
Response: As suggested, we have revised the title and replaced antioxidant capacity with antioxidant activity in the title and throughout the manuscript (line 3, 107).
Comment 2: Also change me-tabolites with metabolites
Response: Needful done and the word ‘me-tabolite’ has been changed to ‘metabolites’ (line 2).
Comment 3: No positive control was presented for antioxidant activity.
Response: Ascorbic acid and Trolox were taken as positive control for ACW and ACL, respectively (line 187).
Comment 4: Please provide the correlation between antioxidant activity and metabolites to show the chemical markers
Response: Antioxidant activity was estimated in a cumulative sample and not in individual fractions of phenolic markers, hence correlation cannot be worked out.
Comment 5: The authors need to mention why they did targeted metabolomics study
Response: We have mentioned in the manuscript (line 75) that there is no information on comprehensive profiling of secondary metabolites at different harvest maturity stages and growing climate in Kinnow. Further, we have mentioned (line 93) that in this paper we are reporting this information which would also complement the published work on primary metabolites.
Comment 6: Pg 1, Line 21-22; There was a sharp increase in the concentration of phenolics, limonin and nomilin at mid-maturity stage which coincided with prevailing low temperature and frost events. Please revise the sentence by mentioning the phenolics that were increased. We cannot see all phenolics are significantly higher in mid maturation stage.
Response: As suggested, we have mentioned the names of polyphenolics (hesperidin, naringin, narirutin, naringenin, neoeriocitrin, rutin, nobiletin and tangeretin) in the manuscript (line 22-23).
Comment 7: Also the same observation is noted in loading plot. Some phenolics are not contributing to the mid stage. Please revise the result and discuss properly as well as justify it.
Response: PCA score plots and loading plots were plotted against all the secondary metabolites identified in Kinnow.
As suggested, we have revised the results and discussed the polyphenolics contributing to higher concentration at mid-maturity stage in the results and discussion section (line 244, 263, 283).
Comment 8: Pg 1, Line 36-39; Citrus fruit have drawn attention of re-36 searchers due to the presence of metabolites with plentiful health benefits such as antiox-idant activity [1,2], anti-proliferative activity [3–7], hypocholesterolaemic and anti-diabetic activity [8–11], anti-inflammatory activity [12–17], and antiretroviral activity [18–20]. Please revise the sentence and avoid the repetition of activity
Response: As suggested, we have revised the sentence and the repetition of word ‘activity’ has been omitted (line 39-41).
Comment 9: Pg 1, line 41; Accumulation of these metabolites in citrus is affected by various factors such as species, variety, climatic and soil conditions of a location, rootstock, and development and maturation. Please revise the sentence (many and) and replace climatic with climate
Response: As suggested, we have revised the sentence (Line 43, page 1) and replaced the word ‘climatic’ with ‘climate’.
Comment 10: Pg 2, line 79; please remove position or check the sentence for suitable word choice
Response: As suggested, we have rewritten the sentence with suitable word choice (line 82).
Comment 11: Please revise the statistical analysis in Table 1. The alphabets are not correctly represented. You have 2 factors. It is better to represent each one of them. Maybe you can put capital and letters. Measurements were made in triplicates (n = 3; 20 Kinnows/replicate). It is confusing. Please revise it
Response: I would like to state that the representation is correct as we have done a two-way ANOVA and given only the interaction effects (C x M) in the tables. Comparison among M within each C and among M between C can be made from the same table looking at the alphabet grouping.
Regarding n=3; 20 Kinnows/replicate, there were three biological replicates used for analysis, therefore, n=3. However, one replicate constituted 20 Kinnows, hence, 20 Kinnows/replicate.
Comment 12: The Figures are not clear and are too small, especially 3, 4 and 5. Please make them clear and increase their resolutions. Also check the statistics for figures.
Response: All the figures have been reworked for better quality and resolution. Statistics for figures is correct as we have done a two-way ANOVA and given only the interaction effects (C x M) in the tables. Comparison among M within each C and among M between C can be made from the same table looking at the alphabet grouping.
Comment 13: The titles of figures should be revised. The labelling is presented in the figure. No need to repeat it in legend
Response: As suggested, we have revised the titles for figures and removed labelling from legends to avoid repetition (line 383, 386, 399).
Comment 14: The conclusion should be rewritten to show the significant findings and answer the objective of the study. please conclude which stage and climate are better
Response: As suggested, we have rewritten the conclusion to show significant findings and included suitable climate and maturity stage in the manuscript (Line 407, 411, 414).
Round 2
Reviewer 2 Report
Dear authors. The revision is satisfactory. However the English is still not improved. For example in line 263; However, a sharp was increase observed in the concentration of some flavonoids such as hesperidin, naringin, narirutin, naringenin, neoeriocitrin, rutin, nobiletin and tangeretin at mid-maturity stage (M3) in Kinnow which coincided with prevailing low temperature and frost events. Please revise the sentence and English throughout the manuscript carefully.
Author Response
The manuscript's english has been improved and revised as suggested.